# Development and Validation of a Questionnaire to Assess Knowledge, Threat and Coping Appraisal, and Intention to Practice Healthy Behaviors Related to Non-Communicable Diseases in the Thai Population

**DOI:** 10.3390/bs7020020

**Published:** 2017-04-14

**Authors:** Kanittha Chamroonsawasdi, Suthat Chottanapund, Pravich Tunyasitthisundhorn, Nawaphan Phokaewsuksa, Tassanee Ruksujarit, Pawarat Phasuksathaporn

**Affiliations:** 1Department of Family Health, Faculty of Public Health, Mahidol University, Bangkok 10400, Thailand; 2Bamrasnaradura Infectious Disease Institute, Ministry of Public Health, Nontaburi 11000, Thailand; 3Medical Research Network of the consortium of Thai medical schools, Bangkok 10900, Thailand; tpravich@hotmail.com; 4Lam Fha Pha Health Promoting Hospital, Samutprakan 10290, Thailand; keawpun2016@gmail.com; 5Ni Klong Bangplakod Health Promoting Hospital, Samutprakan 10290, Thailand; AEMAIMON@hotmail.com; 6Phra Samutchedi Public Health office, Samutprakan 10290, Thailand; pwr_rut@hotmail.com

**Keywords:** NCDs, PMT, reliability test, validation of questionnaire

## Abstract

Non-communicable diseases (NCDs) are important issues in Thailand and health sectors are now focusing on modifiable risks that include cognitive, affective and behavioral factors. This study aimed to develop and validate a questionnaire to assess knowledge about NCDs, threat appraisal, coping appraisal and intention to practice based on protection motivation theory. Content validity was determined by the mean of the item content validity index (I-CVI) from five experts. The questionnaire was pilot tested for difficulty of knowledge items and reliability test using the Kuder-Richardson (KR)-20 and Cronbach’s alpha coefficient among 30 Thai adult subjects in the health office for two sub-districts. The mean I-CVI ranged from 0.90–1.00 and difficulty of knowledge ranged from 0.3–0.9. The reliability test of knowledge by KR-20 ranged from 0.648–0.799, while Cronbach’s alpha coefficients of threat and coping appraisal and intention to practice ranged from 0.70–0.843. We compared sociodemographic data, knowledge about NCDs, threat appraisal, coping appraisal and intention to practice between 50 diabetic type 2 cases and 50 controls. T2DM cases had higher age, knowledge scores on diabetes and hypertension, threat appraisal scores on hypertension and cardiovascular disease when compared with control (*p* < 0.05). The questionnaire was valid and sufficiently reliable to use for data collection.

## 1. Introduction

The World Health Organization (WHO) defines non-communicable diseases (NCDs) as chronic diseases that are not communicable between people with a long duration of illness and generally slow progression [1,2]. There are four common types of NCDs: cardiovascular diseases (hypertension, heart attacks and stroke), cancers (colorectal and lung cancer), chronic respiratory diseases (chronic obstructed pulmonary disease and asthma) and diabetes mellitus (DM) [3]. The NCD Country Profiles of Thailand in 2014 reported that the percentage of deaths among Thai people were 29% cardiovascular diseases (CVD), 17% cancers, 9% chronic respiratory diseases and 4% DM [1]. The WHO and United Nations proposed a prevention and control model for NCDs called a 4 × 4 model [4]. The 4 × 4 model comprises of four risk behaviors: tobacco use, physical inactivity, unhealthy diet and the harmful use of alcohol; and four abnormal physiology markers include hyperglycemia, hyperlipidemia, hypertension; and overweight and obesity [2,3,4,5]. 

According to an increasing number of Thai NCDs patients, methods of modifying risk behavior to prevent NCDs in the Thai population is required. This report presents the preliminary testing of the survey tool for NCDs risk in the Thai population. We developed and constructed a survey questionnaire comprised of 3 parts: subject profile, standard questionnaire for smoking behaviors; excessive alcohol consumption and physical exercise; and cognitive-behavioral factors. In this report, we developed and validated the cognitive-behavioral factors based on the Protection Motivation Theory (PMT) proposed by Rogers [6]. PMT was selected as it is a widely accepted model in health research. The questionnaire comprised of NCD knowledge, threat appraisal, coping appraisal, intention to change behaviors and health related behaviors. When a person has sufficient information and knowledge about NCDs, their cognitive ability can influence an adaptive or maladaptive response to NCDs. To achieve this objective, a valid and reliable instrument was constructed to assess knowledge, threat appraisal and coping appraisal on NCDS, including the intention to practice healthy behaviors in the next six months. Therefore, testing the validity and reliability of the instrument was necessary prior to data collection. 

## 2. Materials and Methods

### 2.1. Conceptual Framework 

In Figure 1, PMT was applied in this study to identify important modifiable risks of NCDs including social cognitive domain comprised of knowledge on NCDs, affective domain comprised of threat appraisal and coping appraisal, and behavioral domain of practicing healthy behaviors. The PMT model was originally applied from the health belief model (HBM) proposed in 1975 [7] to provide conceptual clarity for understanding fear appeals. PMT was first developed within the framework of fear arousing communication that could influence cognition, attitude, behavioral intention and the health behaviors of people regarding exercise, diet, smoking, binge drinking and sexual behaviors [8]. Fear arousal is a drive that motivates trial and error behaviors. In this study, PMT was defined as cognitive processes to describe adaptive-maladaptive coping with a health threat as a result of two appraisal processes: threat appraisal and coping appraisal. Threat appraisal is a process used to evaluate the chance of contracting a disease (perceived vulnerability) and the chance of developing severe illness (perceived severity) after contracting. Coping appraisal is the process that evaluates the components of an individual’s expectation that implementing recommendations can remove the health threat (response efficacy) and the belief in one’s ability to perform coping behaviors by changing health behaviors (self-efficacy). Among NCDs, the health threats are usually focused on physical inactivity, unhealthy diets, cigarette smoking, excessive alcohol consumption and undue stress. The intention to practice is a result of threat appraisal and coping appraisal as it can predict the change of an individual’s health behaviors. When an intention is initiated and continues to be practiced, then the adaptive response to perform recommended behaviors is halted [9]. At present, the intention to practice has been assumed to measure self-reported behaviors as an outcome variable of protection motivation result [10].

### 2.2. Initial Questionnaire Construction

The research team conducted a literature review of knowledge, perceptions and practices covering five NCDs from textbooks and previous studies to construct a questionnaire based on the PMT model [7].

#### 2.2.1. Knowledge on NCDs

The knowledge questions in previous studies usually use two to three categories of answers, i.e., “true/yes” or “false/no”, and some studies have also included “uncertain/unsure” as other answers. The answers were scored as follows: a correct answer = 1, wrong answer and unsure = 0. Five questions for each NCD were constructed with three categories of answers: yes, no and uncertain. Knowledge on diabetic mellitus type 2 (T2DM) comprised common risk factors such as heredity and high sugar intake, symptoms such as frequent hunger, eating too much and body weight loss, and level of fasting blood sugar for diabetics [11,12,13]. Knowledge on hypertension comprised blood pressure classification and risk factors for hypertension such as cigarette smoking, alcohol consumption, obesity and hyperlipidemia [14,15]. Knowledge about cardiovascular disease encompassed risk factors such as stress levels, common symptoms such as severe headache and angina pain [16,17]. Knowledge about colorectal cancer consisted of symptoms such as constipation and bloody stool, and risk factors such as a low fiber diet [18]. Knowledge about chronic obstructive pulmonary disease (COPD) and lung cancer comprised risk factors such as cigarette smoking and heredity, and common symptoms such as difficulty in breathing and restlessness [19,20]. 

#### 2.2.2. Threat Appraisal on NCDs

Threat appraisal comprised perceived vulnerability and perceived severity. From other studies, perception questions about vulnerability and severity of NCDs were classified in a three to five point Likert scale, i.e., strongly agree (5), agree (4), uncertain (3), disagree (2), and strongly disagree (1) or from agree (3), uncertain (2), and disagree (1).

Ten questions of threat appraisal for each NCD were constructed using a three point Likert scale. Perceived vulnerability to T2DM comprised five statements such as “always eating sweet fruit will develop T2DM” and “increased body weight will increase the chance of developing T2DM” [21]. Perceived severity on T2DM comprised five statements, including “chronic renal failure is an important complication of T2DM” and “retinopathy is affected by chronic T2DM” [22].

Perceived vulnerability to hypertension (HT) consisted of five statements, e.g., “smoking cigarettes will develop HT”, “physical inactivity will enhance the risk of HT” and “ex-drinkers may develop more HT” [23,24]. Perceived severity on HT comprised five positive statements, for example, “hypertensive patients will die sooner” and “HT is a risk of cerebrovascular disease” [25].

Perceived vulnerability to CVD comprised five statements, for instance, “chronic stress induces CVD” and “T2DM and HT will increase the risk of CVD” [26,27]. Perceived severity of CVD comprised five statements such as “a sudden death from heart failure is usually found among CVD patients” and “a consequence of cerebrovascular disease was paralysis” [27].

Perceived vulnerability to colon cancer comprised five statements such as “eating processed meat will induce colon cancer” and “eating junk food like French fries, pizza and fried chicken will induce colon cancer” [28,29]. Perceived severity of colon cancer comprised five statements such as “patients with colon cancer will lose intestinal blood” [30].

Perceived vulnerability to COPD and lung cancer comprised five statements such as “living in the same household with smokers will increase the risk of lung cancer” and “patients with COPD are at risk for pneumonitis” [31]. Perceived severity of COPD and lung cancer comprised five statements, for example, “heart failure is a complication of COPD” and “COPD will induce fatigue, poor appetite and weight loss” [32,33].

#### 2.2.3. Coping Appraisal toward NCDs

Regarding coping appraisal toward NCDs, 12 statements about response efficacy were classified in six domains of health behaviors with two statements in each domain addressing healthy food consumption, quitting smoking, quitting alcohol consumption, physical exercise, stress management and self-healthcare [34,35].

Similarly, 35 statements about self-efficacy were classified in six domains of health behaviors, as in response efficacy. Ten statements focused on healthy food consumption; five statements were concerned with quitting smoking and keeping away from smoking zones; five statements focused on quitting alcohol consumption, or avoiding drinking parties; five statements focused on physical activity and active living; five statements were concerned with emotional control and stress management; and five statements focused on self-healthcare [35,36].

#### 2.2.4. Intention to Practice

Ten statements were concerned with the intention to practice healthy behaviors within the next six months based on the PMT and included plans not to smoke [23,34], to not consume alcohol [36], to maintain a healthy diet [36], and have regular physical activity [36].

### 2.3. Subjects and Data Collection Procedures

During the initial phase, the questionnaire was sent to five experts in the field of NCDs for content validity. The item-content validity index (I-CVI) of each question was calculated for content validity and improper statements were revised if the I-CVI was less than 0.5. The revised questionnaire was sent back to the same experts for rechecking.

The finalized questionnaire was tested with T2DM and healthy control groups in two sub-districts under the responsibility of one district health office. The sample size of 50 per group was calculated according to the two-proportion sample formula. A total of 50 diabetic mellitus type 2 (T2DM) cases and 50 healthy subjects living in the same area were recruited as the subjects of the study after informed consents. Data were collected by interviewing the subjects in both groups. 

A reliability test of the instrument was conducted among 20 T2DM cases and 10 healthy subjects. Additionally, demographic data, the mean scores of knowledge, threat appraisal and coping appraisal, and intention to practice healthy behaviors were compared between 50 T2DM cases and 50 healthy subjects. 

### 2.4. Data Analysis

Face content validity and appropriateness of each item in the questionnaire was analyzed. The mean item content validity index (Mean I-CVI) was analyzed using averaged scores of the index of item-objective congruence (IOC) from all experts in each part of the questionnaire. When any question scored an IOC less than 0.5, the question was revised and re-evaluated. Difficulty in the knowledge items was analyzed by the proportion of correct answers by item. When the proportion of correct answers was between 0.2–0.8, the items were neither too difficult nor too easy, and were considered appropriate.

A reliability test was examined from each part of the questionnaire. The Kuder-Richardson-20 (KR-20) was used to identify the reliability test of knowledge while Cronbach’s alpha coefficient was used to test the reliability of threat appraisal, coping appraisal and intention to practice healthy behaviors. A comparison of baseline data, knowledge scores, threat appraisal scores, coping appraisal scores and intention to practice scores between cases and controls were performed by the χ^2^ test and the Mann Whitney U-test. The level of statistical significance was set at *p* < 0.05.

### 2.5. Ethical Consideration

This project was a preliminary study to validate the questionnaire to assess the NCDs risk factors among the Thai population by region. All patients who enrolled in the study submitted a written informed consent. In addition, these patients received all information and explanations on the study objectives and the rights of the patient before enrolling in the pilot study. The data were collected after receiving approval from the ethics committee of the main project.

## 3. Results

The results of the preliminary study to verify the validity test of the questionnaire found that all experts did not suggest removing any item from the questionnaire. The mean item CVI of each section was very high as follows: NCD knowledge = 0.947, threat appraisal = 1.000, coping appraisal = 0.915 and intention to practice healthy behaviors = 0.900 (Table 1).

The difficulty of each item of knowledge regarding T2DM ranged between 0.54–0.80; HT ranged from 0.35–0.77; CVD ranged from 0.34–0.70; colon cancer ranged from 0.57–0.90; and COPD and lung cancer ranged from 0.32–0.94. Regarding the difficulty of knowledge about colon cancer, only the first question stating “the cause of colon cancer was eating grilled meat” was easy to score as the proportion of correct answers was 0.9. Similar to the difficulty of COPD and lung cancer knowledge, one question regarding the cause of COPD and lung cancer was cigarette smoking was easy to score as the proportion of correct answers was 0.94 (Table 2).

The reliability test results on NCD knowledge of each part were at good levels from 0.747–0.799, except that knowledge about COPD and lung cancer was at the lowest. The results of KR-20 on NCDs knowledge can be summarized as follows: T2DM = 0.793, HT = 0.799, CVD = 0.786, colon cancer = 0.747, and COPD and lung cancer = 0.648 (Table 3).

The reliability test results of threat appraisal, comprising perceived vulnerability and perceived severity, ranged from 0.706–0.843 of Cronbach’s alpha coefficient for each NCD threat appraisal was T2DM = 0.706, HT = 0.816, CVD = 0.827, Colon cancer = 0.843, and COPD and lung cancer = 0.817 (Table 3).

Regarding coping appraisal, comprising response efficacy and self-efficacy, the reliability test of response efficacy was equal to 0.805 while self-efficacy was equal to 0.799 (Table 3). The reliability test of intention to practice healthy behaviors was equal to 0.729 (Table 3).

Baseline characteristics between the 50 cases and controls were compared (Table 4). Sex, body mass index and education did not significantly differ between the two groups. The mean average age in the case group was significantly higher than in the control group (57.94 + 9.61 years and 52.0 + 8.35 years, *p*-value = 0.002).

The median scores of knowledge about T2DM and HT among the cases were significant higher than the controls (4.5 with IQR = 2.0 vs. 4.0 with IQR = 3.0, *p* = 0.003 and 4.0 with IQR = 2.0 vs. 3.0 with IQR = 2.75, *p* = 0.005). The other median scores of knowledge about CVD, colon cancer and COPD and lung cancer did not differ between the cases and controls. The median scores of threat appraisal toward HT and CVD among the cases were significantly higher than the controls (28.0 with IQR = 4.0 vs. 27.0 with IQR = 3.0, *p* = 0.021 and 29.0 with IQR = 4.0 vs. 27.5 with IQR = 4.75, *p* = 0.032). The median scores of threat appraisal toward T2DM, colon cancer and COPD and lung cancer between the cases and controls did not significantly differ. The median scores of coping appraisal comprising response efficacy and self- efficacy did not significantly differ between the cases and controls. The median scores of intention to practice also did not significantly differ between the cases and controls (Table 5).

## 4. Discussion

This study aimed to assess the content validity, difficulty of knowledge statement, reliability test of knowledge, threat appraisal, coping appraisal and intention to practice health behaviors prior to its use to collect data among five major NCDs and the control group. The results demonstrated an extremely high value of mean I-CVI in each part of the questionnaire (Mean I-CVI = 0.90–1.00). Larsson et al. [37] constructed a valid and reliable questionnaire with a high I-CVI of 0.8 and over. It also produced good to excellent reliability test results of 0.648–0.799 for Kuder-Richardson-20 concerning the knowledge part and 0.706–0.843 for Cronbach’s alpha coefficient regarding threat appraisal, coping appraisal and intention to practice. When considering the difficulty of knowledge statement, the results also designated the optimum statements to use with calculated difficulty ranging from 0.32–0.9. The knowledge statements were designed to respond to three categories of answers as true, false and uncertain. The results were similar to the study conducted by Najimi et al. [38] on the development and validation of questionnaires to assess knowledge, perception and performance toward obesity with three categories of answers about knowledge and a three point Likert scale for threat and coping appraisal. The same study[38] also had a Cronbach’s alpha coefficient range from 0.5–0.8. This questionnaire was constructed based on the PMT model and was similar to the study conducted by Plotnikoff et al. [10] to identify knowledge about physical activity (PA). This included threat appraisal as perceived vulnerability and severity of PA to induce NCDs, coping appraisal as response efficacy and self-efficacy for performing regular PA, and behavioral intention. In Reference [10], the Cronbach’s alpha coefficients ranged from 0.78–0.9. The PMT was also meaningful in constructing the questionnaire on threat and coping appraisal in predicting behavioral intention [24]. 

When considering the differences between the T2DM cases and control group, it seemed that chronically ill patients had significantly higher knowledge scores about NCDs when compared with the control group. When patients came to healthcare services for treatment, they always received information from their healthcare providers to raise their awareness and practice coping with their illness [11]. NCD risk factors knowledge was linked to risk perception and health-related consequences that increased behavioral change [39].

Among the T2DM cases, the score of threat appraisal toward NCDs such as HT and CVD were significantly higher than those of healthy participants. This is supported by Abed, et al. [27], who reporting perceived vulnerability to HT and acute myocardial infarction as higher among NCDs patients.

## 5. Strengths and Limitations of the Study

The key strengths in the success of this study were the experts’ opinions and the diligence of the local healthcare staff who conducted the NCDs research. The development and validation of the questionnaire took six months prior to testing as all areas, including language, were carefully checked during its development. The pilot questionnaire testing was completed within one month, due to the excellent networks of the local Thai healthcare system with their patients and the willingness of staff. One major limitation is with the model itself. PMT explains the behavior in limited domains and the constructed questions could not represent the whole behavior of the patients. The questionnaire based on the PMT model is quite a long questionnaire, which may result in long interview times in the future. As there were only a limited number of subjects in this pilot, the results may not be representative of the overall Thai population. A large-scale survey would significantly improve the results of the study.

## 6. Conclusions

The validation of the instrument for data collection by identifying content validity, difficulty, and reliability is the most important step to ensure a standard tool for data collection. From our findings, the questionnaire we developed based on the PMT model was suitable and sufficiently valid to use as a tool for data collection on NCD risk factors among the Thai population by region with the results obtained from both experts’ viewpoints and pilot testing in the field with similar characteristics of patients and controls as planned in this project.

## Figures and Tables

**Figure 1 behavsci-07-00020-f001:**
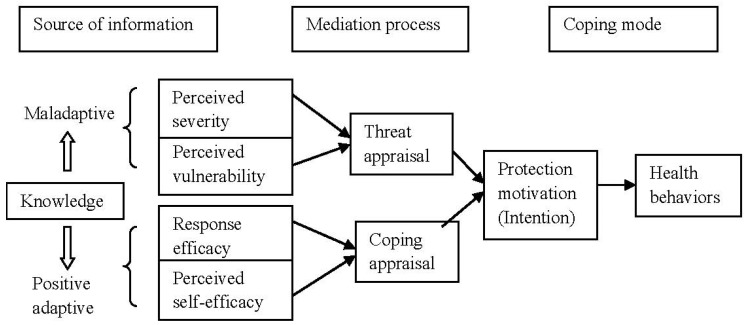
Conceptual framework modified from protection motivation theory of Rogers [6].

**Table 1 behavsci-07-00020-t001:** Mean item content validity index (CVI) of questionnaire.

Part of Questionnaire	No. of Items	Mean Item CVI
NCD knowledge	25	0.947
Threat appraisal	50	1.000
Coping appraisal	47	0.915
Intention to practice	10	0.900

Abbreviations: NCD = non-communicable disease; CVI = content validity index.

**Table 2 behavsci-07-00020-t002:** Difficulty of knowledge questions by disease.

Disease	No. of Items	Difficulty
T2DM Knowledge	5	0.54–0.80
HT knowledge	5	0.35–0.77
CVD knowledge	5	0.34–0.70
Colon cancer knowledge	5	0.57–0.90
COPD and lung cancer knowledge	5	0.32–0.94

Abbreviations: T2DM = diabetes miletus; HT = hypertension; CVD = cardiovascular disease; COPD = chronic obstructive pulmonary disease.

**Table 3 behavsci-07-00020-t003:** Reliability test of questions on NCD knowledge, threat appraisal and coping appraisal.

Part of Questionnaire	No. of Items	Reliability Test
*Knowledge **		
-T2DM.	5	0.793
-HT	5	0.799
-CVD	5	0.786
-Colon cancer	5	0.747
-COPD and Lung cancer	5	0.648
*Threat appraisal (severity & vulnerability) ***		
-T2DM.	10	0.706
-HT	10	0.816
-CVD	10	0.824
-Colon cancer	10	0.843
-COPD and Lung cancer	10	0.817
*Coping appraisal (response efficacy & self-efficacy) ***		
-Response efficacy	12	0.805
-Self efficacy	35	0.799
*Intention to practice health behaviors*	10	0.729

Abbreviations: NCD = non-communicable diseases; T2DM = diabetes miletus type 2; HT = hypertension; CVD = cardiovascular disease; COPD = chronic obstructive pulmonary disease; * using KR-20; ** using alpha coefficient of Cronbach.

**Table 4 behavsci-07-00020-t004:** Baseline characteristics between T2DM cases and controls.

Characteristics	T2DM Cases (*n* = 50)	Controls (*n* = 50)	*p*-Value
Age in years ($\bar{x}\pm SD)$	57.94 ± 9.61	52.0 ± 8.359	0.002 *
Female	70%	76%	0.457
Body mass index ($\bar{x}\pm SD)$	27.05 ± 9.16	25.18 ± 3.07	0.407
Primary school education	70%	50%	0.281

Abbreviations: T2DM = diabetes miletus type 2.

**Table 5 behavsci-07-00020-t005:** Comparison of knowledge, threat appraisal and coping appraisal between T2DM cases and controls.

Scores	T2DM cases (*n* = 50)	Controls (*n* = 50)	*p*-Value
*Knowledge scores*	Median (IQR)	Median (IQR)	
-T2DM	4.5 (2.0)	4.0 (3.0)	0.003 *
-HT	4.0 (2.0)	3.0 (2.75)	0.005 *
-CVD	3.0 (4.0)	3.0 (3.0)	0.158
-Colon cancer	3.0 (2.75)	4.0 (3.0)	0.739
-COPD and Lung cancer	3.0 (3.0)	3.0 (2.0)	0.779
*Threat appraisal scores*			
-T2DM	28.0 (3.0)	28.0 (4.75)	0.105
-HT	28.0 (4.0)	27.0 (3.0)	0.021 *
-CVD	29.0 (4.0)	27.5 (4.75)	0.032 *
-Colon cancer	27.0 (6.0)	26.0 (6.0)	0.085
-COPD & Lung cancer	29.0 (4.0)	29.0 (3.0)	0.383
*Coping appraisal scores*			
-Response efficacy	35.5 (4.0)	36.0 (2.0)	0.357
-Self efficacy	86.5 (14.5)	88.0 (12.75)	0.597
*Intention to practice scores*	30.0(0)	30.0 (1.0)	0.270

Abbreviations: T2DM = diabetes miletus type 2; HT = hypertension; CVD = cardiovascular disease; COPD = chronic obstructive pulmonary disease; IQR = inter quartile range.

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
