# Peer review of "Development and Validation of a Questionnaire to Assess Knowledge, Threat and Coping Appraisal, and Intention to Practice Healthy Behaviors Related to Non-Communicable Diseases in the Thai Population"

_behavsci, 2017, doi:10.3390/bs7020020_

Round 1

Reviewer 1 Report

This work constitutes an important contribution to the literature regarding assessing knowledge, threat and coping appraisal and intention to practice healthy behaviors related to non-communicable diseases (NCDs) in the Thai population. Such tools are singularly important to myself and my international teams with expertise in lifestyle-related NCDs and working to make these the least causes of premature death in the world rather than the leading causes. Of concern is the growing rate of children with risk factors. Knowledge of the dimensions reflected in the validated questionnaire in this study will help inform cross cultural adaptations to addressing these priority conditions. Although the idea per se is not highly original it is singularly important. The rationale is well developed and cogent. The methods are generally sound and are theory/model based. Having said that I do recommend that the sampling frame and recruitment process be described in greater detail. How was a diagnosis of presumably type 2 diabetes mellitus established? Were individuals with type 1 diabetes mellitus included? How was the sample sizes of the two groups established? In addition, I recommend securing ethical approval from the University of the investigators even though this is after the fact. The results are adequately and clearly displayed (abbreviations need to be addressed, see below) and the discussion is appropriate. The primary limitation of this work is that is does not conform to journal style and format (although the English language is quite good however there are some areas for improvement). Most of my comments are editorial. In addition to conforming to journal style and format: Replace 'toward' in the title with 'related to' Minimize the number of abbreviations. CA xx is not acceptable. When abbreviations are used, include the abbreviation in the first instance of the word in the manuscript Replace 'toward' in the title with 'related to' See journal policy re person-first language; better to refer to participants with and without type 2 diabetes mellitus (T2DM) Refer to T2DM throughout vs. DM Knowledge 'about' vs. 'on' In most instances, risk factors 'for' vs. 'of' (but 'of' appropriate in a few instances) In English, sentences should not begin with an abbreviation by convention; usually can insert 'The' before the abbreviation. Check throughout for parallel construction particularly in relation to lifestyle behaviors, e.g., Line 112, 'excess alcohol consumption and undue stress' Line 68. Edit to 'sedentary behavior' Line 72. Edit to 'time being sedentary over 24 hours' With respect to alcohol consumption as in Line 81, better to include 'excess' alcohol consumption Line 82 ''premature' deaths Line 85. Edit to 'women and men.' Line 93. Edit to 'NCDs' Line 94. Edit to 'within the next' Line 99. Edit to 'adapted from' Fig 1. Unless required by the journal, insert all figures and tables at the end of the manuscript, each on a separate page. Makes for easier reviewing. Line 134. Edit to 'knowledge about' vs. 'on', throughout the Ms. Legend of Fig 1. Insert 'the' before 'PMT here and elsewhere, however in the legend, best to have PMT in full Line 148. Delete 'in' Line 155. Generally the preposition after 'vulnerability is 'to' Line 159 and elsewhere. Preferable to spell out HT in the interest of reducing excessive abbreviations. As mentioned, CA colon, CA xx, is too informal. Spell out colon cancer, etc. Preferable to refer to participants vs. patients or subjects throughout. Line 181. Include a sentence at the end of this paragraph beginning 'Health responsiveness is defined as.........(references). I am in the field of health protection and health promotion, and I do not know what 'health responsiveness' means. Line 185. Be careful with the use of words throughout. Do you really mean quitting alcohol consumption or reducing to avoid abusive or excessive consumption? If an alcoholic then clearly quitting is the goal. However, most people who consume alcohol consume it responsibly. I believe the emphasis is not drinking excessively. Line 221. Ethical consideration. Edit the first line to 'This project constituted a preliminary study on the assessment of...in the Thai population...' In the Discussion, the format of the reference citations is not typical, i.e., the inclusion of investigators' initials. Edit accordingly. Line 270. Edit to 'This study aimed to assess the ....' Line 298. Change the construction as described elsewhere to 'Among those participants with T2DM...' Line 306. Edit to '...questionnaire constructed and based on the PMI was suitable and sufficiently valid to use...' References. Check journal style. Check abbreviations for all journals.

Author Response

Response to Reviewer #1

#1.This work constitutes an important contribution to the literature regarding assessing knowledge, threat and coping appraisal and intention to practice healthy behaviors related to non-communicable diseases (NCDs) in the Thai population. Such tools are singularly important to myself and my international teams with expertise in lifestyle-related NCDs and working to make these the least causes of premature death in the world rather than the leading causes. Of concern is the growing rate of children with risk factors. Knowledge of the dimensions reflected in the validated questionnaire in this study will help inform cross cultural adaptations to addressing these priority conditions. Although the idea per se is not highly original it is singularly important. The rationale is well developed and cogent. The methods are generally sound and are theory/model based.

[SC] Thank you very much for a very positive comment on our manuscript.

#2.Having said that I do recommend that the sampling frame and recruitment process be described in greater detail. How was a diagnosis of presumably type 2 diabetes mellitus established? Were individuals with type 1 diabetes mellitus included? How was the sample sizes of the two groups established? In addition, I recommend securing ethical approval from the University of the investigators even though this is after the fact.

[SC] The method part has been revised as below.

2.3. Subjects and data collection procedures

At the initial phase, the constructed questionnaire was sent to five experts in the field of NCDs for content validity.  After checking each part for content validity and revising improper statements the revised questionnaire was rechecked by the experts.  

The finished questionnaire was tested with T2DM and healthy control in provincial health office. The sample size of 50 per group was calculated according to two-proportion sample formula. Total of 50 Diabetic Mellitus type 2 (T2DM) cases and 50 healthy subjects living in the same area (two subdistricts in one province) were recruited as the subjects of study after informed consents.  Data were collected by interviewing the subjects in both groups.  

A reliability test of the instrument was conducted among 20 T2DM cases and 10 healthy subjects.  Additionally, demographic data, mean scores of knowledge, threat appraisal and coping appraisal and intention to practice healthy behaviors were compared between 50 T2DM cases and 50 healthy subjects

#3.The results are adequately and clearly displayed (abbreviations need to be addressed, see below) and the discussion is appropriate. The primary limitation of this work is that is does not conform to journal style and format (although the English language is quite good however there are some areas for improvement).

[SC] All abbreviations and English correction has been done by a native English speaker.

#4.Most of my comments are editorial. In addition to conforming to journal style and format: Replace 'toward' in the title with 'related to' Minimize the number of abbreviations. CA xx is not acceptable. When abbreviations are used, include the abbreviation in the first instance of the word in the manuscript Replace 'toward' in the title with 'related to' See journal policy re person-first language; better to refer to participants with and without type 2 diabetes mellitus (T2DM) Refer to T2DM throughout vs. DM Knowledge 'about' vs. 'on' In most instances, risk factors 'for' vs. 'of' (but 'of' appropriate in a few instances) In English, sentences should not begin with an abbreviation by convention; usually can insert 'The' before the abbreviation. Check throughout for parallel construction particularly in relation to lifestyle behaviors, e.g., Line 112, 'excess alcohol consumption and undue stress' Line 68. Edit to 'sedentary behavior' Line 72. Edit to 'time being sedentary over 24 hours' With respect to alcohol consumption as in Line 81, better to include 'excess' alcohol consumption Line 82 ''premature' deaths Line 85. Edit to 'women and men.' Line 93. Edit to 'NCDs' Line 94. Edit to 'within the next' Line 99. Edit to 'adapted from' Fig 1. Unless required by the journal, insert all figures and tables at the end of the manuscript, each on a separate page. Makes for easier reviewing. Line 134. Edit to 'knowledge about' vs. 'on', throughout the Ms. Legend of Fig 1. Insert 'the' before 'PMT here and elsewhere, however in the legend, best to have PMT in full Line 148. Delete 'in' Line 155. Generally the preposition after 'vulnerability is 'to' Line 159 and elsewhere. Preferable to spell out HT in the interest of reducing excessive abbreviations. As mentioned, CA colon, CA xx, is too informal. Spell out colon cancer, etc. Preferable to refer to participants vs. patients or subjects throughout. Line 181. Include a sentence at the end of this paragraph beginning 'Health responsiveness is defined as.........(references). I am in the field of health protection and health promotion, and I do not know what 'health responsiveness' means. Line 185. Be careful with the use of words throughout. Do you really mean quitting alcohol consumption or reducing to avoid abusive or excessive consumption? If an alcoholic then clearly quitting is the goal. However, most people who consume alcohol consume it responsibly. I believe the emphasis is not drinking excessively. Line 221. Ethical consideration. Edit the first line to 'This project constituted a preliminary study on the assessment of...in the Thai population...' In the Discussion, the format of the reference citations is not typical, i.e., the inclusion of investigators' initials. Edit accordingly. Line 270. Edit to 'This study aimed to assess the ....' Line 298. Change the construction as described elsewhere to 'Among those participants with T2DM...' Line 306. Edit to '...questionnaire constructed and based on the PMI was suitable and sufficiently valid to use...'

[SC] All mentioned texts have been changed.

#5.References. Check journal style. Check abbreviations for all journals.

                [SC] The references have been checked and corrected.

Reviewer 2 Report

To the author

This study is important. The research is novel and addresses non-communicable disease risks that can be used to inform its mitigation or prevention in Thailand. However, the paper is rather difficult to fully understand and we have some suggestions below.

Abstract

·         More information on who the questionnaire was developed and validated for would be good (i.e. Thai adults not mentioned). This has implications for your current conclusions and the external validity of these findings to the general Thai population.

Introduction

·         The substantial detail in the introduction, while interesting, is unnecessary for this paper and distracts from the principle focus which must be on the validation itself.

·         Consider reducing the material presented in the first four paragraphs to just one introductory paragraph.

·         The rest of the introduction should focus on the purpose of this paper – why was it done, what was the problem being addressed, and what was the approach taken overall.

·         How do the authors justify using PMT theory to structure their survey instrument? This theory is 35 years old and health-behaviourists have developed many other theories that need to be acknowledged and valued in the context of this research.

Methods

·         The overall impression is of a complex exercise competently attempted by an expert group. However, for a non-psychometrician described procedures are hard to follow.

·         Accordingly, it would help non-experts if the authors were to prepare a flow-chart showing the steps taken from the input from the panel of experts to the various components measuring knowledge, threat appraisal, coping appraisal, and intention to practise healthy behaviours.

·         Describe and justify the cases and controls, their selection, and their number.

·         Questionnaire construction-more information on your search would be good

·         Were items that were used for this questionnaire standardized in previous populations?

Sampling

·         How were experts sampled to determine inter-rater bias?

·         how many cases-controls were approached (only 100 or did some decline participation?) was this a convenience sample? -implications for external validity

·         How was the smaller sample size determined for the reliability testing and why were more ‘cases’ sampled than ‘non-cases’ (a bit more detail would be helpful)? And why was this restricted to DM

Data analysis

·         Perhaps separate ‘face content validity’ to face validity and content validity and say that both types of validity testing were used in combination (if this is true)

·         Why are some data components referred to as ‘baseline data’?

·         Describe the statistical tests used (Chi squared, Mann Whitney-U, I-CVI, Cronbach’s alpha, and Kuder-Richardson-20 (KR-20)) and indicate clearly the utility and application of each test.

·         Was S-CVI tested to assess inter-rater agreement?

Discussion

·         Can you provide some discussion about why your results have come out similar to previous studies? Are the populations comparable and/or are the questions used really good at measuring the outlined constructs?

·         What are the implications of your findings and how will this inform your future work?

·         Please include strengths and limitations-see below some ideas

For example strengths could include:

·         use of multiple sources to construct your survey may make this survey stronger at identifying multiple diseases and reduce redundant survey taking for participants

·          the strength of the evidence that the items were selected from

·         Relative brevity and ease of administration of the developed survey instrument (consider including the instrument as an appendix)

·          

For example limitations could include:

·       Generalizability and external validation

·         Implications of only including DM for reliability testing when future work will be carried out with multiple NCDs

·         Discuss the impact that these findings might have for other groups (selection bias?)

·         Will these findings be reproducible given the size and nature of sample?

·         Are there potential limitations of content and face validity that should be discussed (i.e. high subjectivity of experts and or differences between expert opinions)? 

Author Response

Response to Reviewer #2

#1.This study is important. The research is novel and addresses non-communicable disease risks that can be used to inform its mitigation or prevention in Thailand. However, the paper is rather difficult to fully understand and we have some suggestions below.

                [SC] Thank you for very useful comments and very positive perspective toward our manuscript.

                 #2.Abstract

·         More information on who the questionnaire was developed and validated for would be good (i.e. Thai adults not mentioned). This has implications for your current conclusions and the external validity of these findings to the general Thai population.

[SC] The abstract has been changed as comment.

                  #3.Introduction

·         The substantial detail in the introduction, while interesting, is unnecessary for this paper and distracts from the principle focus which must be on the validation itself.

·         Consider reducing the material presented in the first four paragraphs to just one introductory paragraph.

·         The rest of the introduction should focus on the purpose of this paper – why was it done, what was the problem being addressed, and what was the approach taken overall.

·         How do the authors justify using PMT theory to structure their survey instrument? This theory is 35 years old and health-behaviourists have developed many other theories that need to be acknowledged and valued in the context of this research.

                [SC] The introductory paragraph has been changed to one paragraph as comment.

                    #4.Methods

·         The overall impression is of a complex exercise competently attempted by an expert group. However, for a non-psychometrician described procedures are hard to follow.

·         Accordingly, it would help non-experts if the authors were to prepare a flow-chart showing the steps taken from the input from the panel of experts to the various components measuring knowledge, threat appraisal, coping appraisal, and intention to practise healthy behaviours.

·         Describe and justify the cases and controls, their selection, and their number.

·         Questionnaire construction-more information on your search would be good

·         Were items that were used for this questionnaire standardized in previous populations?

                [SC] The method has been explained as comment.

                      #5.Sampling

·         How were experts sampled to determine inter-rater bias?

·         how many cases-controls were approached (only 100 or did some decline participation?) was this a convenience sample? -implications for external validity

·         How was the smaller sample size determined for the reliability testing and why were more ‘cases’ sampled than ‘non-cases’ (a bit more detail would be helpful)? And why was this restricted to DM

Data analysis

·         Perhaps separate ‘face content validity’ to face validity and content validity and say that both types of validity testing were used in combination (if this is true)

[SC] The sampling method has been explained as comment.

·                      #6.Why are some data components referred to as ‘baseline data’?

·         Describe the statistical tests used (Chi squared, Mann Whitney-U, I-CVI, Cronbach’s alpha, and Kuder-Richardson-20 (KR-20)) and indicate clearly the utility and application of each test.

·         Was S-CVI tested to assess inter-rater agreement?

Discussion

                [SC] The questionnaire construction method has been added as comment.

·                      #7.Can you provide some discussion about why your results have come out similar to previous studies? Are the populations comparable and/or are the questions used really good at measuring the outlined constructs?

·         What are the implications of your findings and how will this inform your future work?

·         Please include strengths and limitations-see below some ideas

For example strengths could include:

·         use of multiple sources to construct your survey may make this survey stronger at identifying multiple diseases and reduce redundant survey taking for participants

·          the strength of the evidence that the items were selected from

·         Relative brevity and ease of administration of the developed survey instrument (consider including the instrument as an appendix)

·          For example limitations could include:

·       Generalizability and external validation

·         Implications of only including DM for reliability testing when future work will be carried out with multiple NCDs

·         Discuss the impact that these findings might have for other groups (selection bias?)

·         Will these findings be reproducible given the size and nature of sample?

·         Are there potential limitations of content and face validity that should be discussed (i.e. high subjectivity of experts and or differences between expert opinions)?

                [SC] The discussion of above comments has been added in the manuscript.

Round 2

Reviewer 1 Report

The work has merit and offers new knowledge, and appears carefully done. As non-native English speakers, the authors/investigators have made their very best effort regarding the writing of the manuscript. To achieve the level of English writing (grammar and composition) as well as technical style and adherence to scientific writing and journal style requirements, I recommend that the manuscript be proof read and edited by a native English speaker with exceptional grammar and composition knowledge. In addition, the copy editor needs familiarity with scientific writing and journal style standards (note the tables should be on separate pages; the formatting consistent, e.g., use of capitalization or not within tables needs to be consistent). The references need to be checked line by line. At present, many journal names are not appropriately capitalized.

Author Response

Thank you for your very positive comment. The manuscript was language edited by MDPI langauge service. The certificate was attached. 
